*Resource*

# A functional assay for serum detection of antibodies against SARS-CoV-2 nucleoprotein

Anna Albecka[1] ID, Dean Clift[1] ID, Marina Vaysburd[1], Tyler Rhinesmith[1], Sarah L Caddy[1,2] ID, David M Favara[3,4,5], Helen E Baxendale[6] & Leo C James[1,*] ID

## Abstract

**The humoral immune response to SARS-CoV-2 results in antibodies against spike (S) and nucleoprotein (N). However, whilst there are widely available neutralization assays for S antibodies, there is no assay for N-antibody activity. Here, we present a simple *in vitro* method called EDNA (electroporated-antibody-dependent neutralization assay) that provides a quantitative measure of N-antibody activity in unpurified serum from SARS-CoV-2 convalescents. We show that N antibodies neutralize SARS-CoV-2 intracellularly and cell-autonomously but require the cytosolic Fc receptor TRIM21. Using EDNA, we show that low N-antibody titres can be neutralizing, whilst some convalescents possess serum with high titres but weak activity. N-antibody and N-specific T-cell activity correlates within individuals, suggesting N antibodies may protect against SARS-CoV-2 by promoting antigen presentation. This work highlights the potential benefits of N-based vaccines and provides an *in vitro* assay to allow the antibodies they induce to be tested.**

**Keywords** antibodies; neutralization; nucleoprotein; SARS-CoV-2; TRIM21
**Subject Category** Microbiology, Virology & Host Pathogen Interaction
**The EMBO Journal (2021) 40: e108588**

## Introduction

The immune response to SARS-CoV-2 results in a strong antibody response to both spike (S) and nucleoprotein (N) (Li *et al*, 2020). Antibodies against these two antigens have been widely used as diagnostics of past or present SARS-CoV-2 infection. Anti-S antibodies (or S antibodies) have been shown to neutralize SARS-CoV-2 and inhibit its replication, leading to their use as a rapid and effective antiviral treatment. Research into S antibodies has also played a vital role in the development of S-based vaccines, both by demonstrating that raising S antibodies is a worthwhile objective and by

providing neutralization assays that allow correlates of protection to be established, a vital measure of vaccine efficacy. In contrast, despite the fact that anti-N antibodies (or N antibodies) are found in SARS-CoV-2 convalescents at levels that equal or exceed those of S antibodies (Rydyznski Moderbacher *et al*, 2020), there has been comparatively less investigation into their relevance beyond diagnostics. This is largely because unlike with anti-S antibodies, there is no assay that can measure if N antibodies block SARS-CoV-2 infection. However, it seems unlikely that a robust anti-N response plays no role in immunity to SARS-CoV-2. Studies in mice using murine hepatitis virus (MHV; a murine β-coronavirus with brain and liver tropism) have shown that passively transferred N antibodies are protective (Nakanaga *et al*, 1986; Lecomte *et al*, 1987). Indeed antibodies to internal antigens like N have been shown to prevent infection by arenaviruses (Richter & Oxenius, 2013; Straub *et al*, 2013), ebolavirus (Wilson *et al*, 2000), human cytomegalovirus (HCMV; Bootz *et al*, 2017), human immunodeficiency virus (HIV; Excler *et al*, 2014; Mayr *et al*, 2017), influenza viruses (Sambhara *et al*, 2001; Carragher *et al*, 2008; LaMere *et al*, 2011) and vaccinia virus (Moss, 2011). Crucially however, because internal antigens are usually hidden inside the virion, antibodies against them do not bind infectious viral particles. Consequently, N antibodies and similar typically do not block infectious entry of viruses into cells in standard *in vitro* assays and are described as "non-neutralizing". The mechanisms behind the immune protection provided by non-neutralizing antibodies like N antibodies remain largely unknown.

Importantly, without functional data on SARS-CoV-2 N antibodies or an available assay to measure protective activity, there is less incentive to invest significant global effort into developing N-based vaccines. Meanwhile, although S-based vaccines are proving highly effective against SARS-CoV-2 transmission, there is growing evidence that emerging spike variants of SARS-CoV-2 are less susceptible to the immunity they induce (preprint: Garcia-Beltran *et al*, 2021; preprint: McCallum *et al*, 2021). The emergence of spike variants has also been seen in patients treated with convalescent plasma/monoclonals (Avanzato *et al*, 2020). The observed rapid

1  MRC Laboratory of Molecular Biology, Protein & Nucleic Acid Division, Cambridge, UK
2  CITIID, Department of Medicine, University of Cambridge, Cambridge, UK
3  Department of Oncology, Addenbrooke's Hospital, Cambridge University Hospitals NHS Foundation Trust, Cambridge, UK
4  Department of Oncology, The Queen Elizabeth Hospital, The Queen Elizabeth Hospital King's Lynn NHS Foundation Trust, Kings Lynn, UK
5  Department of Oncology, University of Cambridge, Cambridge, UK
6  Royal Papworth NHS Trust, Cambridge, UK
  *Corresponding author. Tel: +44 1223 267162; E-mail: lcj@mrc-lmb.cam.ac.uk

mutation of SARS-CoV-2 spike underlies the widely held view that new vaccines may be required seasonally, unless the virus can be globally eradicated. One solution is to keep making new variant S-vaccines, but another is to use a combined approach immunizing with a vaccine that contains both S and N, making it much less likely for a resistant virus to emerge. All 10 currently authorized and approved SARS-CoV-2 vaccines are S-based. We therefore set out to develop an assay for N-antibody activity, both to provide evidence for the inclusion of N as a candidate vaccine antigen and to allow N-based vaccines to be efficiently tested. The assay we developed, named EDNA (for electroporated-antibody-dependent neutralization assay), provides the only *in vitro* method that allows the antiviral activity of N antibodies in SARS-CoV-2 convalescent serum to be rapidly and quantitatively tested.

## Results

### Antibodies mediate intracellular neutralization of MHV

As MHV is a class 1 biosafety pathogen, and N antibodies are known to play a role in immunity to MHV *in vivo* (Nakanaga *et al*, 1986; Lecomte *et al*, 1987), we decided to use this as a model system to develop an assay that could be used to measure N-antibody activity. First, we established a system to quantify MHV replication based on syncytia formation and cytopathic effect (CPE). Live phase-contrast microscopy revealed cell–cell fusion occurs 8–10 h after MHV-A59 infection and peaks at 30 h, followed by lysis and cell death (Movie EV1, Appendix Fig S1A). Plotting either cell area quantified from live imaging or cell viability determined by total ATP levels at 48 h post-infection against virus dilution gave almost identical dose–response curves and TCID50 values (Appendix Fig S1B–D). Thus, live-cell imaging can be used to follow cytopathic MHV infection over time and accurately quantify virus titre. Using this approach, we tested the neutralization capacity of a polyclonal antiserum raised against disintegrated, purified MHV-A59 virions (Rottier *et al*, 1981) that includes antibodies against both MHV-A59 S and N (Appendix Fig S1E). Antiserum was added to the cell media, or delivered directly into the cytosol by adapting our previously described Trim-Away technology (Clift *et al*, 2017). In Trim-Away, antibodies are electroporated into cells and form a complex with their protein target. This complex is recognized by the cytosolic Fc receptor and E3 ubiquitin ligase TRIM21 (James *et al*, 2007), which uses ubiquitination to recruit the proteasome and mediate complex degradation (Mallery *et al*, 2010; Kiss *et al*, 2021). Remarkably, the antiserum reduced MHV-A59 replication 10-fold more potently when delivered intracellularly than added extracellularly (Fig 1A–C). This suggests that there are antibodies present in the antiserum that can bind to viral proteins post-fusion in the cytosol and neutralize replication.

To test if anti-N antibodies are responsible for the block to replication, we electroporated L929 cells with serial dilutions of an anti-N monoclonal antibody (Leibowitz *et al*, 1987) or a control anti-GFP antibody and infected with MHV-A59. Strikingly, electroporation of the anti-N monoclonal antibody completely neutralized MHV-A59, whereas the control anti-GFP antibody had no effect (Fig 1D–F). Previously, we have shown that intracellular antibody-dependent neutralization (ADIN), in which non-enveloped viruses

pre-bound by antibody are neutralized after entry into the cytosol, is dependent on TRIM21 (Mallery *et al*, 2010). Upon binding to antibody-coated viruses, TRIM21 mediates rapid proteasomal- and VCP-dependent viral degradation (Hauler *et al*, 2012) and thereby blocks replication. To test whether this is also the case for electroporated anti-N antibodies and MHV-A59, we generated CRISPR TRIM21 knockout L929 cells (Appendix Fig S1F). TRIM21 knockout did not affect cell proliferation or cytopathic MHV-A59 infection in the absence of serum or antibody (Appendix Fig S1G). However, TRIM21 knockout cells could no longer neutralize MHV-A59 when electroporated with MHV antiserum or anti-N monoclonal antibody, suggesting that intracellular neutralization is completely TRIM21-dependent (Fig 1G and H). To further confirm the role of TRIM21 in intracellular neutralization, we co-electroporated anti-N antibody with excess recombinant protein A/G, which competes with TRIM21 for binding to IgG Fc (Keeble *et al*, 2008). The presence of protein A/G did not impact antibody electroporation efficiency or localization in the cytosol (Appendix Fig S1H and I). However, co-electroporation of protein A/G with anti-N antibody completely abolished intracellular neutralization of MHV-A59 in otherwise wildtype cells (Fig 1H). These data suggest that anti-N antibodies neutralize MHV-A59 intracellularly by recruiting the cytosolic Fc receptor TRIM21. In both ADIN and Trim-Away, TRIM21 exerts its effects by targeting protein:antibody complexes for degradation. We therefore tested whether electroporated anti-N antibodies are causing degradation of MHV-A59 N protein. To do this, we allowed N-protein expression for 4 h post-infection, blocked further N-protein expression by addition of cycloheximide (CHX) and then electroporated anti-N or anti-GFP antibodies or a no antibody control. Electroporation in the absence of antibody did not change N-protein levels (Appendix Fig S1J). Electroporation of anti-N but not anti-GFP antibody led to a reduction in N-protein levels (Fig 1I). This is consistent with the block to MHV replication being caused by N-protein degradation. Taken together, the data show that our electroporation-based method can be used to measure the activity of N antibodies *in vitro*. To distinguish this from the natural process of antibody-dependent intracellular neutralization (ADIN), we designated the approach "electroporated-antibody-dependent neutralization assay" (EDNA).

### EDNA can be used to measure intracellular antibody neutralization of SARS-CoV-2

Next, we investigated whether EDNA could be used to investigate the N-antibody response to SARS-CoV2. To do this, we used a clinical isolate of SARS-CoV-2/human/Liverpool/REMRQ0001/2020 and Vero cells modified to stably express SARS-CoV-2 entry factors ACE2 and TMPRSS2 (Papa *et al*, 2021). We added polyclonal rabbit anti-N antibodies or control rabbit IgG to Vero ACE2/TMPRSS2 with or without electroporation and after 24 h infected them with SARS-CoV-2. To assess virus replication, we lysed the cells 24 h after infection and checked levels of genomic viral RNA by RT–qPCR. Electroporation of N antibodies reduced viral RNA by 3-logs (Fig 2A). Importantly, the same sera in the absence of electroporation had no impact on replication, indicating that N antibodies must be inside the cell to mediate neutralization. To confirm these results, we measured the effect of anti-N sera on the production of infectious particles after 24 h by plaque assay. In agreement with RT–qPCR,

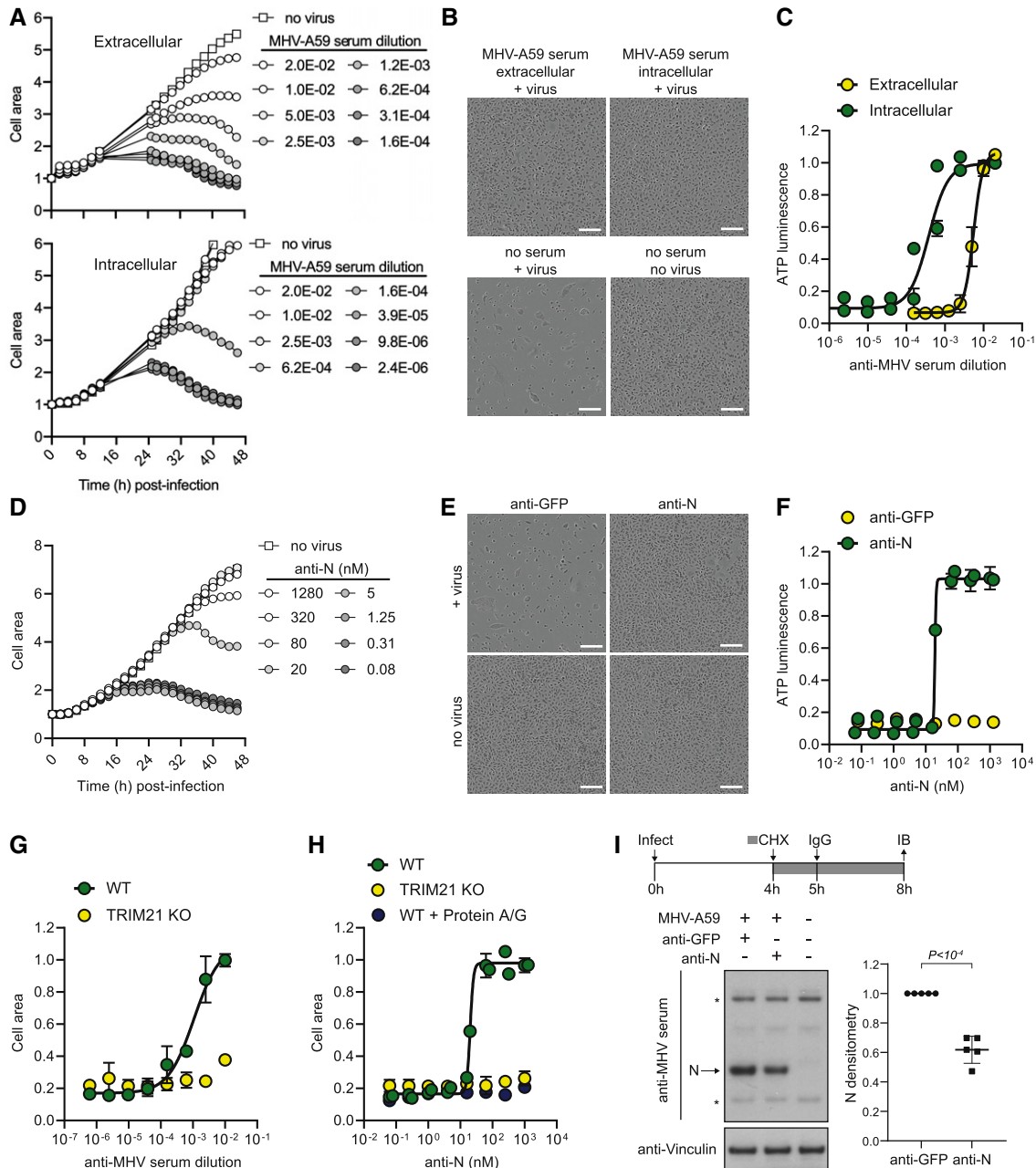

**Figure 1. Electroporated-antibody-dependent neutralization assay (EDNA) for coronavirus.**

A   A titration of anti-MHV polyclonal serum was added either directly to media (extracellular) or electroporated into L929 cells (intracellular), and then, cells were infected with MHV-A59. The kinetics of cell growth was monitored by measuring total cell area.

B   Phase-contrast images of cells at 48 h post-infection. Scale bar = 200 μm.

C   At 48 h post-infection, the viability of cells infected with MHV-A59 in the presence of intracellular or extracellular antiserum was determined by ATP luminescence assay. Increasing doses of antiserum results in increased cell survival.

D   Kinetics of cell growth following MHV-A59 infection in the presence of a titration of electroporated anti-N antibody.

E   Phase-contrast images of cells at 48 h post-infection. Scale bar = 200 μm.

F   Quantification of cell viability by ATP luminescence assay at 48 h post-infection in the presence of electroporated anti-N or anti-GFP antibodies.

G, H   WT or TRIM21 KO L929 cells were infected with MHV-A59 in the presence of electroporated polyclonal antiserum (G) or anti-N antibody (H) and quantified by cell area 48 h later. Co-electroporation of protein A/G with anti-N antibody into WT cells mimics the TRIM21 KO phenotype.

I   L929 cells were infected with MHV-A59 for 4 h, and then, cyclohexamide (CHX) was added to block further viral protein synthesis. After 5 h, cells were electroporated with anti-N or anti-GFP antibodies and left for a further 3 h before being western blotted for cellular N protein levels (* denotes a non-specific band).

Data information: Data were analysed using a Student's *t*-test. Error bars depict the mean +/- SEM. All data represent at least two independent replicates.
Source data are available online for this figure.

we observed a significant reduction in production of new infectious particles in the presence of electroporated N antibodies (Fig 2B). Intracellular neutralization was antibody dose-dependent, consistent with the MHV data and previous ADIN experiments (McEwan *et al*, 2012) (Fig 2C and D). We next investigated whether intracellular neutralization of SARS-CoV-2 is TRIM21-dependent. We transduced a previously characterized TRIM21 knockout HEK293T clone with ACE2, to make it sensitive to SARS-CoV-2 entry, and then reconstituted TRIM21 expression with lentivirus encoding TRIM21 under its endogenous promoter or transduced with an empty vector control (Fig 2E; Zeng *et al*, 2019). Electroporation/infection experiments were then carried out as before in both cell lines. Neutralization of SARS-CoV-2 by N-antibody electroporation was observed by both

RT–qPCR (Fig 2F) and plaque assay (Fig 2G) but only in cells reconstituted with TRIM21. These results confirm that N-specific antibodies mediate TRIM21-dependent neutralization of SARS-CoV-2 and that EDNA can be used to quantify this activity.

### EDNA provides a functional assay for N antibodies in SARS-CoV-2 convalescent sera

Anti-N antibodies reach high titres in many seroconverted individuals (preprint: Hachim *et al*, 2020) and are widely used as a diagnostic for previous infection with SARS-CoV-2 (Krammer & Simon, 2020). However, there is no equivalent to an S-antibody neutralization test for N. We therefore investigated whether EDNA could

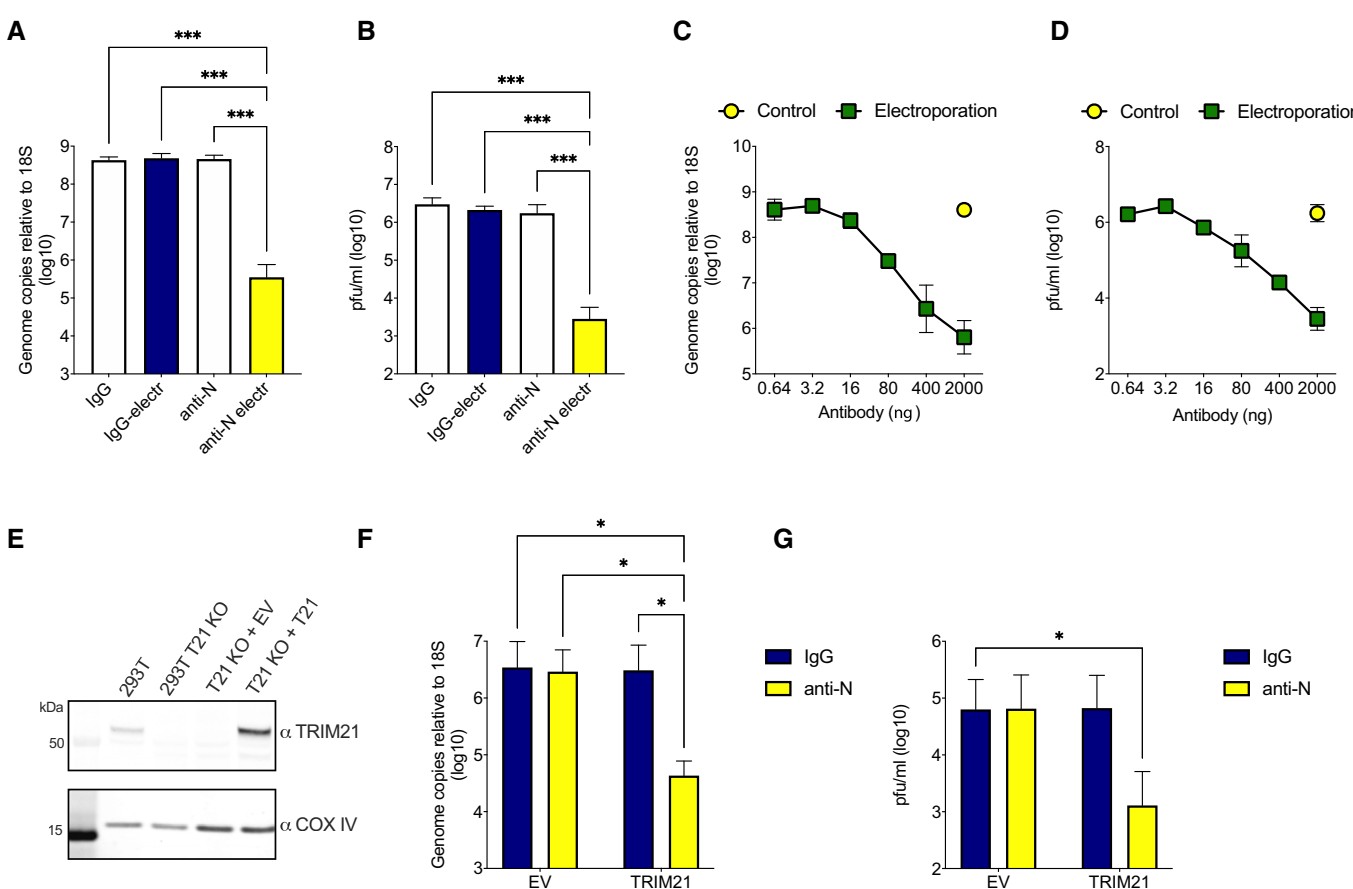

**Figure 2. N antibodies inhibit SARS-CoV-2 replication intracellularly.**

A, B   Vero cells OE ACE2 and TMPRSS2 were infected with SARS-CoV-2 in the presence of IgG or anti-N antibodies added directly into media or electroporated into cells. Viral replication was then determined by RT–qPCR (A) or plaque assay (B). Electroporation of anti-N antibodies significantly inhibits SARS-CoV-2 replication (***$P < 0.0002$).

C, D   As with A&B, except with a titration of electroporated anti-N antibodies.

E   Western blot of 293T cells and 293T ACE2 OE/TRIM21 KOs alone or reconstituted with empty vector (EV) or an endogenous promoter-driven TRIM21 vector (T21) (Zeng *et al*, 2019).

F, G   293T ACE2 OE/TRIM21 KO cells reconstituted with EV or TRIM21, electroporated with IgG or anti-N antibodies, and infected with SARS-CoV-2. Viral replication was then determined by RT–qPCR (F) or plaque assay (G). Electroporation of anti-N antibodies significantly inhibits SARS-CoV-2 replication only in TRIM21-reconstituted cells (*$P < 0.05$).

Data information: All data represent at least three independent replicates. Error bars depict the mean ± SEM. Statistical comparisons were performed using a one-way (A, B) or two-way (F, G) ANOVA.
Source data are available online for this figure.

provide such an assay. We tested a small cohort of four SARS-CoV-2 convalescents (SARS-CoV-2 seropositivity confirmed by luminex testing) and two sero-negatives for their antibody responses using a capillary-based protein detection system (Jess, ProteinSimple). Jess measures native antibody–antigen binding, meaning that quantification of the chemiluminescent signal provides a measure analogous to the antibody titre obtained by ELISA (i.e. it is a combination of specific antibody concentration and affinity). We observed a range of responses; some sera displayed only limited reactivity to any SARS-CoV-2 antigen, others reacted robustly against N but weakly against S, and some had both strong anti-N/strong anti-S profiles (Fig 3A and B). The data show that a good dynamic range

in antigen reactivity is capable of being measured but a larger cohort would be required to draw any conclusions about antibody portfolios. Next, we electroporated the sera into 293T ACE2 cells and infected them 24 h later with SARS-CoV-2. Infection levels were assessed after a further 24 h by plaque assay. Infection was decreased > 10-fold in cells electroporated with sera that had a strong anti-N response (Fig 3C). In contrast, sera with essentially no anti-N antibodies were unable to neutralize infection. To establish whether activity in the polyclonal sera is dependent upon TRIM21, we compared infection in 293T TRIM21 KO ACE2 cells reconstituted with either empty vector or TRIM21. Only in TRIM21-reconstituted cells did electroporation of anti-N sera reduce infection (Fig 3D).

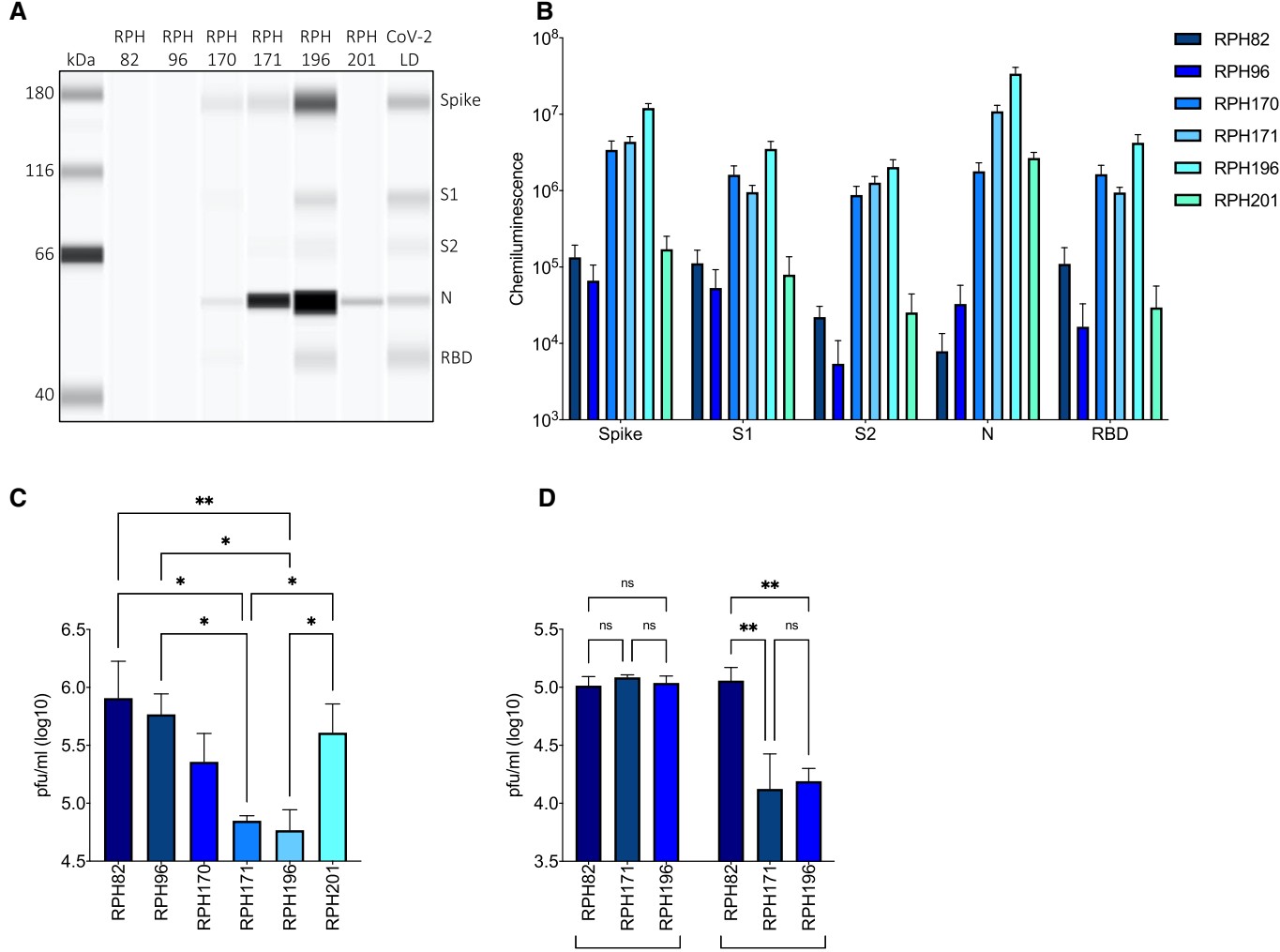

**Figure 3.  EDNA detects intracellular neutralization activity in SARS-CoV-2 convalescent sera.**

A  Capillary-based protein detection (Jess) of antibodies against SARS-CoV-2 antigens in convalescent sera.

B  Quantification of antigen-specific antibodies in convalescent sera.

C  293T cells OE ACE2 infected with SARS-CoV-2 in the presence of electroporated convalescent sera. There are statistically significant differences in the ability of serum from different individuals to inhibit viral replication intracellularly (**$P < 0.005$, *$P < 0.05$).

D  293T ACE2 OE/TRIM21 KOs reconstituted with either empty vector (EV) or TRIM21-expressing vector (TRIM21) infected with SARS-CoV-2 in the presence of electroporated convalescent sera. Intracellular neutralization of viral replication is only observed in cells reconstituted with TRIM21 (**$P < 0.005$).

Data information: All data represent at least three independent replicates. Error bars depict the mean ± SEM. Statistical comparisons were performed using a one-way (C) or two-way (D) ANOVA.

**SARS-CoV-2 convalescents make potently neutralizing anti-N antibodies and this correlates with increased levels of N-specific T cells**

Having validated that EDNA could be used to assess convalescent serum samples, we collected a larger panel of sera from seroconverted staff at Royal Papworth Hospital NHS Foundation Trust (RPH), Cambridge University Hospitals NHS Foundation Trust (CUH) in Cambridge and The Queen Elizabeth Hospital NHS Foundation Trust (QEH) in Kings Lynn, UK. As before, we used Jess to quantify the antibody responses to N, S, S1, S2 and RBD (Fig 4A and Appendix Fig S2). There was a wide range in response strength, with chemiluminescence varying over > 4-logs. Choosing two of the strongest responders to N protein, we performed a titration to establish the linear range of the binding assay (Appendix Fig S3). Sera containing particularly strong responses were diluted and re-measured to be within the linear range. The titration also confirmed that measured weak responders are still well above the threshold for detection. Performing a comparison between the antigens, we observed a weak correlation between anti-N and anti-S, S1, S2 or RBD responses ($R^2$ between 0.17–0.32) (Appendix Fig S4A–D). Nevertheless, there was considerable variation with substantial numbers of individuals possessing monodominant responses (strong N or S but not both).

Next, we electroporated a single dilution of each serum into Vero ACE2/TMPRSS2 cells, challenged them with SARS-CoV-2 and quantified infection by RT–qPCR (Fig 4B). Comparing intracellular neutralization activity with antibody titres to each antigen in turn showed a trend between neutralization potency and strength of response to N but not to other antigens (Fig 4C and Appendix Fig S2F–I). This is consistent with the expected antigen topology upon viral entry, in which S antigen will remain at the plasma membrane following membrane fusion whilst N is delivered into the cytosol with the viral genome. To further confirm that it is anti-N and not anti-S antibodies that are responsible for the observed intracellular neutralization, we tested serum possessing a strong S/weak N antibody response (Appendix Fig S2A, Fig 4A) and confirmed that whilst it is unable to neutralize intracellularly, it potently neutralizes extracellularly (Appendix Fig S4E vs. Fig 4B). Whilst sera with the highest titre anti-N responses was usually the most strongly neutralizing, there were exceptions to this trend. Four serum samples possessed neutralization activity that was unexpectedly high given their modest anti-N titres, whilst one sample had a high anti-N titre but gave modest neutralization (Fig 4A–C; green and red bars, respectively). This may be because antibodies with unusually high affinity but present at low levels can exert a protective effect whilst conversely, specific antibodies present at high concentrations but with weak affinities may not. Comparison of the Jess and neutralization data also suggests that there is a threshold effect, in that neutralization is only observed in sera with a chemiluminescent signal > $10^6$ (Fig 4C). To determine whether this is due to the nature of the antibodies in these samples or whether there is a functional threshold for intracellular neutralization of SARS-CoV-2, we electroporated a titration of two potently neutralizing serum samples. We observed similar neutralization curves for both, indicating that once diluted to a chemiluminescent signal < $10^6$, neutralization activity is largely lost (Fig 4D and Appendix Fig S3). This suggests that there is indeed a functional threshold for intracellular neutralization of SARS-CoV-2.

Antibodies against internal antigens like nucleoprotein are generated during the immune response to most enveloped viruses (Schmaljohn, 2013). Importantly, although anti-N antibodies do not block infection when added directly to cultured cells *in vitro*, they provide protective immunity *in vivo*. This is likely because the antibodies are not efficiently taken up by cells *in vitro*, a step we bypass in EDNA by using electroporation. How anti-N antibodies are imported by cells *in vivo* and provide protection is not well understood. Recently, we showed that anti-N antibodies help to clear LCMV infection by promoting the induction of N-specific cytotoxic T cells (Caddy *et al*, 2021). This is thought to occur as a result of cross-presentation: nucleoprotein immune complexes (N:Ab) are imported by antigen-presenting cells and detected by TRIM21, leading to proteasomal degradation and the generation of N peptides for MHC class I presentation. Both B and T cells responses contribute to SARS-CoV-2 immunity and almost all convalescents have both (Sette & Crotty, 2021). We therefore considered that the same immune mechanism of antibody:T cell synergy we observed during LCMV infection may be operating during the response to SARS-CoV-2. To test this hypothesis, we isolated fresh PBMCs from SARS-CoV-2 convalescents and used ELISpot to quantify their N-specific T cells. Upon stimulation with an N peptide library, we observed a ∼ 100-fold range in the number of specific (interferon expressing) cells between individuals (Fig 4E). Comparing the number of N-specific T cells and intracellular neutralizing activity within each individual revealed a modest correlation, consistent with the hypothesis that N antibodies may contribute to protection against SARS-CoV-2 by promoting T-cell immunity (Fig 4F). Of note, a similar correlation performed using anti-N binding titre failed to give a convincing correlation (Fig 4G). This result highlights the importance of measuring the neutralization activity of anti-N antibodies, not just their levels in serum, and the utility of EDNA.

## Discussion

Here, we have described an *in vitro* assay called EDNA that can quantify the activity of N antibodies produced upon SARS-CoV-2 infection, in the same way that classical neutralization assays are used to quantify S-antibody activity (for overview, see Summary Figure). EDNA allows the measurement of N-antibody activity in SARS-CoV-2 convalescent sera, can be used in conjunction with standard readouts like RT–qPCR and plaque assay and has a large dynamic range. EDNA uses electroporation to deliver antibodies directly into the cytoplasm of cells, before exposing them to virus. This allows antibodies that target antigens normally hidden inside the viral envelope to be tested for their ability to disrupt infection. Importantly, this means that antiviral activity can be detected in antibody sera that on the basis of low anti-S titres would normally be characterized as non-neutralizing (e.g. RPH38, CUH186, CUH066; Fig 4A and B, Appendix Fig S2A–D). Neutralizing antibodies typically work extracellularly by inhibiting receptor binding and/or preventing fusion of the viral envelope with the cell membrane. In EDNA experiments with both MHV and SARS-CoV-2, and with both polyclonal sera and monoclonal anti-N antibodies, we show that intracellular neutralization is dependent upon the cytosolic Fc receptor and E3 ubiquitin ligase TRIM21. TRIM21 has previously been shown to mediate antibody-dependent intracellular

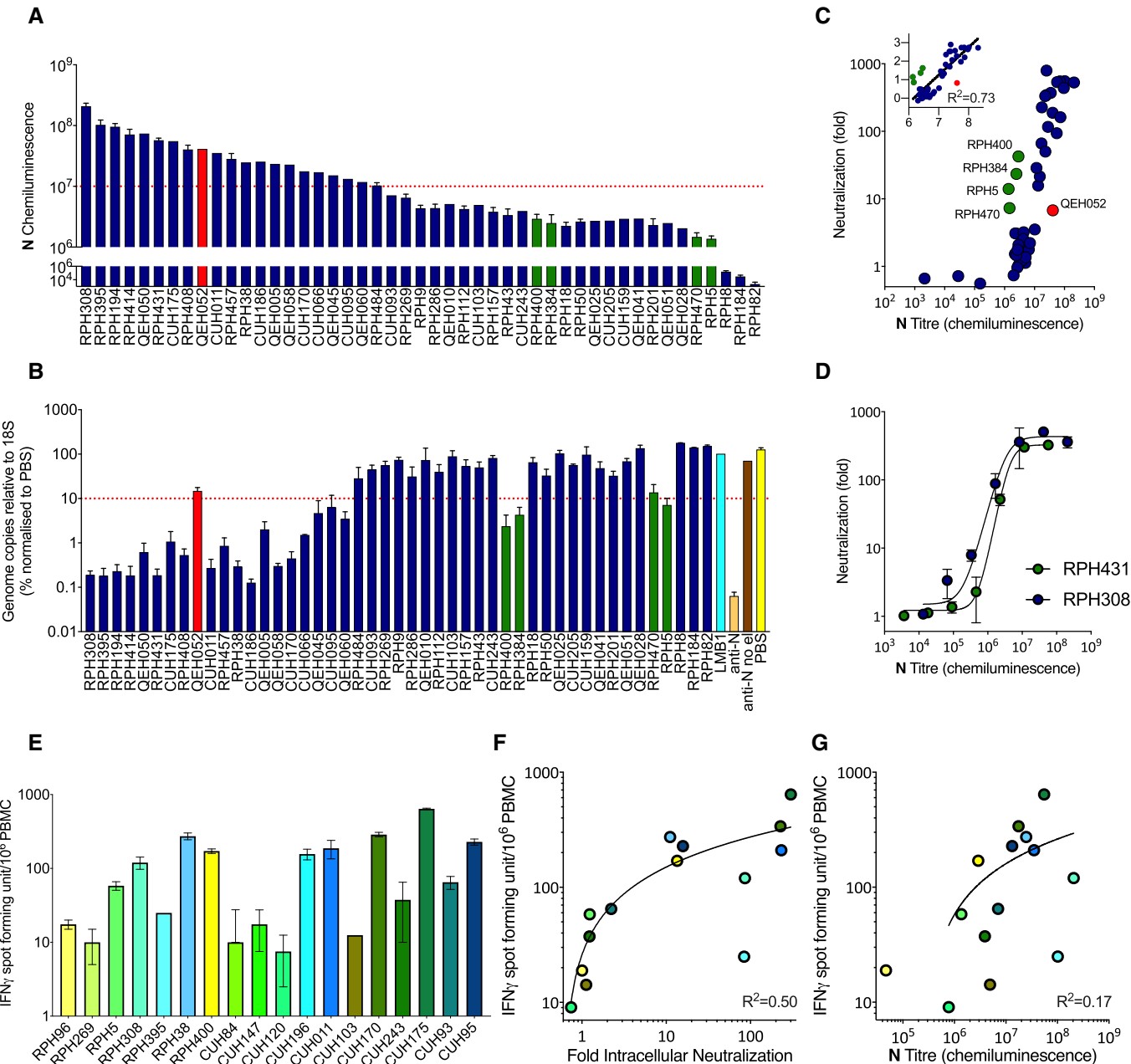

**Figure 4. SARS-CoV-2 convalescents make potently neutralizing N antibodies whose activity correlates with the number of active N-specific T cells.**

A   Capillary-based protein detection (Jess) of N antibodies in convalescent sera, quantified by chemiluminescence.

B   Quantification of SARS-CoV-2 intracellular neutralization by electroporated convalescent sera using EDNA and RT–qPCR (as % of PBS condition).

C   Correlation between N-antibody titre, as calculated by Jess, and intracellular neutralization (fold decrease in SARS-CoV-2 replication), as measured by EDNA, in convalescent sera from multiple individuals.

D   Intracellular neutralization of SARS-CoV-2 by titrated serum from two strongly N antibody-positive individuals.

E   ELISpot measuring IFNγ production in T cells from SARS-CoV-2 convalescents upon stimulation with a peptide library covering N-protein.

F   Correlation between N-specific T-cell activity and intracellular neutralization activity.

G   Correlation between N-specific T-cell activity and N-antibody titre.

Data information: All data represent at least three independent replicates. Error bars depict the mean ± SEM. Statistical analysis was performed using a semilog non-linear fit (F, G).

neutralization (ADIN). In ADIN, antibodies pre-bound to a non-enveloped virus recruit TRIM21 to cause capsid degradation and we hypothesize that a similar mechanism of degradation operating

against N protein blocks viral replication during EDNA. N protein protects the coronavirus genome and stabilizes sub-genomic RNA during viral transcription – so its degradation would be catastrophic

for viral replication. Consistent with this, we observe N protein degradation during EDNA experiments with MHV, whilst in SARS-CoV-2 experiments intracellular neutralization activity correlates with N antibody but not S-antibody titre. N-antibody binding titres are not always predictive of intracellular neutralizing capacity however, as sera can differ in titre by > 10-fold but possess similar neutralization activity (e.g. RPH308 vs. CUH186; Fig 4A and B). Based on previous data, we predict that off-rates between antibody–antigen (Bottermann *et al*, 2016) and antibody-TRIM21 (Foss *et al*, 2016) will be predictive of intracellular neutralization potency. In contrast with extracellular entry-blocking neutralization, we expect epitope specificity to be less important as long as the epitope is present in the folded state and TRIM21 binding can take place.

Infection with most enveloped viruses results in high titre antibodies against internal antigens and whilst these are typically non-neutralizing *in vitro* (at least as characterized in standard assays) they are often highly protective *in vivo* (Schmaljohn, 2013). Based on our results here, and recently published (Caddy *et al*, 2021), we propose that ADIN may provide one underlying mechanism by which non-neutralizing antibodies mediate protection. However, ADIN does not explain how and where non-neutralizing antibodies like those against N protein meet their antigen. EDNA is an *in vitro* assay in which antibodies are artificially delivered directly into the cytosol. This is a limitation of the method and does not address the issue of cellular uptake and cytosolic import during natural SARS-CoV-2 infection *in vivo*. Whilst antibodies are readily taken up by most tissues, they are normally recycled back out of the cell by the neonatal Fc receptor FcRn. There is evidence that antibodies can be imported into the cytosol, typically during disease, but the mechanisms involved are unclear (Congdon *et al*, 2013). Indeed, it is the absence of a well-understood mechanism to explain how antibodies against internal viral antigens like the nucleoprotein meet their target has discouraged study into their role in protective immunity.

The context in which cytosolic antibody import is best understood is during antigen presentation, where both passive mechanisms involving membrane disruption and leakage into the cytosol (Reis e Sousa & Germain, 1995) and active mechanisms of import requiring the channel forming protein Sec61 (Mukai *et al*, 2011) and the ATPase VCP (Ackerman *et al*, 2006) have been proposed. Most recently, the receptor DNGR-1 has been shown to promote phagosomal rupture by inducing NADPH oxidase activity (Canton *et al*, 2021). Previous work on viruses such as influenza has demonstrated a synergistic link between N-specific antibodies and N-specific cytotoxic T lymphocytes (CTLs; LaMere *et al*, 2011; Laidlaw *et al*, 2013). Moreover, we recently showed that TRIM21 uses anti-N antibodies to promote N-specific CTLs and thereby protect against LCMV infection (Caddy *et al*, 2021). These data suggested that TRIM21 can detect immune complexes upon their import into antigen-presenting cells and cause their rapid proteasomal degradation, leading to the efficient generation of peptides for MHC class I presentation. We hypothesize that a similar process may be operating during SARS-CoV-2 infection. Consistent with this, we find a correlation between N-antibody neutralization as measured by EDNA and N-specific T cells as measured by ELISpot. Convalescents with potently neutralizing N antibodies possess increased numbers of N-specific T cells. Moreover, we observed individuals (e.g. RPH38; Fig 4A, B, E and Appendix Fig S2A) with low S titres but both strong anti-N neutralization and high numbers of N-specific T cells. There may be differences in the immediate protection afforded by N- vs S-directed B- and T-cell immunity. Perhaps more importantly, the longevity of responses to each antigen may differ as could their ability to protect against new viral variants. Notably, N-specific T-cell responses in SARS-CoV convalescents could be detected 17 years after infection (Le Bert *et al*, 2020) and were cross-reactive with N from SARS-CoV-2. Whilst our findings require further validation in larger cohorts, and in studies correlating responses with patient outcomes, they provide evidence both of the utility of EDNA and the importance of considering N-based vaccines as a viable alternative to S-only approaches.

# Materials and Methods

## Reagents and Tools table

| Reagent/Resource | Reference or Source | Identifier or catalog number |
|---|---|---|
| **Experimental models** | | |
| Vero CCL-81 | ATCC | CCL-81 |
| HEK293T | ATCC | CRL-3216 |
| L929 | ATCC | CCL-1 |
| Vero ACE2/TMPRSS2 | Papa *et al* (2021) | Vero ACE2/TMPRSS2 |
| HEK293T ACE2 | Papa *et al* (2021) | HEK293T ACE2 |
| SARS-CoV-2/human/Liverpool/REMRQ0001/2020 | Papa *et al* (2021) | Kind gift from Lance Turtle (University of Liverpool) and David Matthews and Andrew Davidson (University of Bristol) |
| MHV-A59 | | Kind gift from Ian Goodfellow (Cambridge University) |
| **Antibodies** | | |
| MHV-A59 polyclonal antiserum raised against disintegrated, purified MHV-A59 virions (Rottier *et al*, 1981) | Dr. Peter Rottier (Utrecht University). | MHV-A59 polyclonal antiserum |

**Reagents and Tools table** (continued)

| Reagent/Resource | Reference or Source | Identifier or catalog number |
|---|---|---|
| Monoclonal Anti-Murine Coronavirus Nucleocapsid (N) Protein, Clone 1.16.1 | BEI Resources, NIAID, NIH | NR-45106 |
| Mouse anti-GFP antibody | Rockland | 9F9.F9; 600-301-215 |
| Rabbit anti-N antibodies | Millipore | ABIN129544 |
| Rabbit normal IgG | Millipore | 12-370 |
| Rabbit anti-TRIM21 | Abcam | ab207728 |
| Mouse TRIM21 | Santa Cruz | sc-21367 |
| Rabbit anti-Vinculin | Abcam | ab217171 |
| Rabbit anti-COX IV | LICOR | 926-42214 |
| HRP-coupled secondary anti-mouse light chain specific | Millipore | AP200P |
| Anti-rabbit light chain specific | Millipore | MAB201P |
| Anti-SARS-CoV-2 S | Invitrogen | PA1-41165 |
| Anti-IFN-γ antibody 1-D1K | Mabtech | cat#3420-3 |
| Anti-biotin monoclonal antibody | Vector Labs | #SP-3020 |
| IgG-A488 | Molecular Probes | A11055 |
| **Oligonucleotides and other sequence-based reagents** | | |
| *For long lists of oligos or other sequences please refer to the relevant Table(s) or EV Table(s)* | | |
| CDC-N2 (IDT 2019-nCoV RUO kit) | CDC | 143503 |
| SARS-CoV-2_N_Positive control RNA | IDT | 10006625 |
| **Chemicals, Enzymes and other reagents** (*e.g. drugs, peptides, recombinant proteins, dyes etc.*) | | |
| Methylene blue | Sigma | M4159 |
| Toluidine blue | Sigma | M4159 |
| Formaldehyde | Sigma | T3260 |
| ATPlite 1-step luminescence reagent | Perkin Elmer | 6016731 |
| RIPA buffer | Cell Signalling Technology | CST-9806 |
| NuPAGE LDS Sample Buffer | Thermo Fisher | NP0007 |
| NuPAGE 4–12% Bis-Tris Gels | Thermo Fisher | NP0326BOX |
| Protease Inhibitor Cocktail | Roche | P8340 |
| ECL Western Blotting Detection Reagent | Amersham | RPN2106 |
| SARS-CoV-2 serology assay | Bio-Techne | SA-001 |
| Separation module 12–240 kDa | Bio-Techne | SM-W004 |
| RNAsecure | Invitrogen | AM7006 |
| Luna® Universal Probe One-Step | NEB | E3006 |
| PepTivator SARS-CoV-2 protein N peptide pool | Miltenyi Biotec | 130-126-698 |
| Lymphoprep | StemCell | 07801 |
| ELISpot MAIP plates | Millipore | MAIPS4510 |
| 8-well chambered coverslips | Ibidi | 41122111 |
| Cas9 protein | IDT | #1081060 |
| Alt-R Cas9 Electroporation Enhancer | IDT | #1075915 |
| **Software** | | |
| Compass | Bio-Techne | https://www.proteinsimple.com/compass/downloads/ |
| Eclipse software | Sony | |
| GraphPad Prism 9 | GraphPad | |
| **Other** | | |
| PMA-Lite LED Photolysis blue light device | Generon | E90002 |
| Jess capillary protein detection system | ProteinSimple | |

**Reagents and Tools table**  (continued)

| Reagent/Resource | Reference or Source | Identifier or catalog number |
|---|---|---|
| Neon Transfection System | Thermo Fisher | MPK5000 |
| IncuCyte S3 | Sartorius | IncuCyte S3 |
| ABI StepOnePlus PCR System | Life Technologies | |
| ELISpot reader | AID ispot, Autoimmun Diagnostika | |
| Eclipse EC800 flow cytometer | Sony | |
| Lumascope LS720 widefield microscope | Etaluma | |
| PHERAstar FS plate reader | BMG | |

## Methods and Protocols

### MHV EDNA

1  Wash cells (e.g. Vero ACE2/TMPRSS2) in PBS and resuspend in Buffer R (Thermo Fisher) at a concentration of between $0.1–1 \times 10^8$ cells $ml^{-1}$ depending on experiment.

2  For each electroporation reaction, mix $0.1–1 \times 10^6$ cells (10.5 µl) with 2 µl of the antibody/serum/protein to be delivered.

3  Take up electroporation mixture into a 10-µl Neon Pipette Tip and electroporate using a Neon Electroporator using the following settings: 1,400 V, 20 ms, 2 pulses. For non-electroporated controls, skip this step.

4  Transfer electroporated cells to medium supplemented with 10% serum without antibiotics.

5  Plate $1 \times 10^4$ electroporated L929 cells in 96-well plates in triplicates. The next day, infect cells with MHV-A59 at MOI = 1.

6  Image using an IncuCyte system for 48 h within a 37°C, 5% $CO_2$ humidified incubator.

7  Quantify total cell area using IncuCyte software by detecting the outline of cells. Mock-infected cells (no virus) will rapidly proliferate to form a confluent monolayer over the course of 48 h (Appendix Fig S1A).

8  Check for a typical cell growth curve consisting of a short lag phase followed by logarithmic growth and then a stationary phase as the cells formed a confluent monolayer. In contrast, cells infected with MHV-A59 should begin to fuse after 8–10 h, and by 30 h post-infection large syncytia occupy the majority of the monolayer. Between 30 and 48 h post-infection, the syncytia should have lysed and detached from the dish and the remaining cells exhibited a rounded morphology typical of dead of dying cells. The cell growth curves should show an incremental shift towards the no virus control with each serial dilution of viral stock (Appendix Fig S1B, circles – black to grey to white). Plotting the cell area values at 48 h post-infection against virus dilution should give a typical dose–response curve that can be fitted with a non-linear regression curve to calculate a TCID50 value (in our case, $2.12 \times 10^8$ TCID50/ml; Appendix Fig S1C).

9  After imaging is completed, the same cells should be analysed by ATP luminescence assay at 48 h post-infection to quantify cell viability.

10  For ATP assay, cells in 100 µl media in dark 96-well plates should be lysed with 100 µl/well of ATPlite 1-step luminescence reagent according to manufacturer instructions.

11  Quantify luminescence using a luminometer (e.g. BMG PHERAstar FS plate reader). Plotting these values against virus dilution should give an almost identical dose–response curve and TCID50 value (Appendix Fig S1D).

### SARS-CoV-2 EDNA

1  Wash cells (e.g. Vero ACE2/TMPRSS2) in PBS and resuspend in Buffer R (Thermo Fisher) at a concentration of between $0.1–1 \times 10^8$ cells $ml^{-1}$ depending on experiment.

2  For each electroporation reaction, mix $0.1–1 \times 10^6$ cells (10.5 µl) with 2 µl of the antibody/serum to be delivered.

3  Take up electroporation mixture into a 10 µl Neon Pipette Tip and electroporate using a Neon Electroporator using the following settings: 1,400 V, 20 ms, 2 pulses. For non-electroporated controls, skip this step.

4  Transfer electroporated cells to medium supplemented with 10% serum without antibiotics.

5  Seed $1.5 \times 10^4$ electroporated cells into 96-well plates in triplicates for RT–qPCRs or $2.5 \times 10^5$ cells into 24-well plates for plaque assays. After 24 h, transfer plates to containment level 3 laboratory (CL3).

6  Remove supernatants and wash wells with PBS to remove remaining antibodies that could interfere with the virus entry process.

7  At CL3, infect cells at moi = 1 in DMEM supplemented with 2% FBS and antibiotics.

8  Assess viral RNA loads or production of viral particles after 24 h incubation to allow for single replication cycle (see below). Plates should be immediately frozen after incubation. For plaque assays, cells should be freeze/thawed three times to ensure virions are released.

9  To compare EDNA with classical extracellular neutralization, incubate sera directly with virus at a range of pfu for 1 h before adding to cells for 24 h. Assess neutralization by following step 8 above.

### Plaque assay for SARS-CoV-2

10-fold serial dilutions of viral supernatants were prepared and used to infect monolayers of Vero ACE2/TMPRSS2 cells. After 1 h of incubation, wells were overlayed with DMEM containing 2% FBS, antibiotics and 0.05% agarose. Cells were incubated for 3 days, fixed with 4% formaldehyde and stained with 0.1% toluidine blue.

### RT–qPCR assay for SARS-CoV-2

After incubation plates were immediately frozen at −70°C to help with cell lysis. Next, plates were thawed at 4°C and 1 volume of lysis buffer (0.25% Triton X-100, 50 mM KCl, 100 mM Tris–HCl pH 7.4, glycerol 40% and RNAsecure (1/100) added to wells and mixed gently by pipetting up and down few times. After 5 min of lysis, cell lysates were transferred to PCR plates and virus inactivated at 95°C for 5 min. RT–qPCR were performed with Luna® Universal Probe One-Step kit following manufacturer recommendations. Primer/probe for genomic viral RNA were CDC-N2. Primer probe for 18S control were described previously (Ashraf *et al*, 2006). SARS-CoV-2_N_Positive control RNA was used as standard for the viral genomic N reactions. For 18S standard, DNA was synthesized and kindly gifted by Jordan Clarks and James Stewart (University of Liverpool). Final concentrations of 500 nM for each primer and 125 nM for the probe were used. RT–qPCRs were run on ABI StepOnePlus PCR System with following program: 55°C for 10 min, 95°C for 1 min and then 40 cycles of 95°C denaturation for 10 s and 60°C extension for 30 s. RNA copy numbers were obtained from standards and then genomic copies of N normalized to $10^{10}$ copies of 18S. Finally, all data were normalized to 100% negative control.

### Generation of TRIM21 KO cells

L929 TRIM21 KO cells were generated using the Alt-R CRISPR-Cas9 system from Integrated DNA technologies (IDT) with a predesigned crRNA sequence (GAGCCTATGAGTATCGAATG). Guide RNA in the form of crRNA-tracrRNA duplex was assembled with recombinant Cas9 protein and electroporated into L929 cells together with Alt-R Cas9 Electroporation Enhancer. Two days post-electroporation polyclonal cells were expanded and also plated one cell per well in 96-well plates to select single cell clones screened by Western blotting for TRIM21 protein.

### SARS-CoV-2 virus preparation

Virus stock was generated in Vero ACE2/TMPRSS2 cells by infecting cells at low moi of 0.05 and incubating for three days. Supernatants were freeze/thawed three times, aliquoted and stored at −70°C. Titres were assessed by plaque assay. MHV-A59 virus was a kind gift from Ian Goodfellow (University of Cambridge).

### Western blotting

Cells were washed in PBS, lysed in RIPA buffer (CST-9806) or 1% Triton X-100 supplemented with a protease inhibitor cocktail, spun at 14,000 *g* for 10 min and cleared lysates mixed with NuPAGE LDS Sample Buffer and heated at 95°C for 5 min. Samples were run on NuPAGE 4–12% Bis-Tris gels and transferred onto nitrocellulose membrane. For Western blotting of MHV-A59, MHV-A59 polyclonal antiserum was used that was raised against disintegrated, purified MHV-A59 virions (Rottier *et al*, 1981). The serum detects multiple viral proteins by immunoblot that likely represent the entire MHV-A59 proteome. Prominent bands at ∼180, ∼90 and ∼50 kDa likely correspond to full-length spike (S), cleaved spike and nucleoprotein (N), respectively (Appendix Fig S1E). Antibody incubations were rabbit anti-MHV serum (1:5,000), mouse anti-MHV N protein (1:2,000), rabbit anti-TRIM21 (1:1,000) or mouse TRIM21 (1:1,000), rabbit anti-Vinculin (1:50,000) and rabbit COX IV antibodies (1:5,000). HRP-coupled secondary anti-mouse light chain-specific and anti-rabbit light chain-specific antibodies were detected by enhanced chemiluminescence. For fluorescent detection, we used secondary antibodies from LICOR and LICOR Odyssey detection device.

### Serum and cell samples

Staff from RPH were recruited through staff email over the course of 2 months: 20 April 2020-10 June 2020, as part of a prospective study to establish seroprevalence and immune correlates of protective immunity to SARS-CoV-2 (Ethics Approval: HRA IRAS: 96194 REC: 12/WA/0148). Staff from CUH and QEH were recruited through the COVID-19 Serology in Oncology Staff (CSOS) study (Favara *et al*, 2021a; Favara *et al*, 2021b) with samples collected in June, July and December 2020 (Ethics approval: HRA IRAS: 284231). Samples were screened for SARS-CoV-2 N and S binding antibodies by Luminex assay as previously described (Kemp *et al*, 2021) and selected for further investigation based on selective binding profiles.

### Serum inactivation

Human serum samples from RPH were inactivated by methylene blue photochemical treatment. For that, methylene blue was added to sera at final concentration of 4 μM and illuminated for 15 min with PMA-Lite LED Photolysis blue light device. Serum samples from CUH and QEH were heat inactivated at 56°C for 30 min.

### Detection and quantification of anti-SARS-CoV-2 antibodies

Inactivated human serum was run on Jess capillary protein detection system. Serum samples were diluted 20× for the first test. Higher or lower dilutions were used for consequent runs depending on antibody signal. To detect S-, S1-, S2-, N- and RBS-specific antibodies, we used SARS-CoV-2 serology assay and separation module 12–240 kDa. Samples were run following manufacturers protocol. Shortly, protein standard is run in each capillary and in the presence of specific human antibodies these serve as primary antibodies that are then detected with anti-human HRP secondary antibody. Bio-Techne software Compass was used to quantify antibody titres in the samples.

### PBMC isolation and ELISpot

PBMCs were isolated from whole blood collected into Lymphoprep. ELISpot plates containing PVDF membranes were activated with 15 μl of 35% ethanol for 30 s and washed with distilled water. Plates were then coated overnight at 4°C with 100 μl of monoclonal antibodies against IFN-γ 5 μg/ml of clone 1-D1K. ELISpot plates were washed and then blocked with 200 μl R-10 media for at least 3 h. R-10 media: RPMI 1640 supplemented with 10% (v/v), FBS, 2 mM ʟ-glutamine, 100 units penicillin, 0.1 mg/ml streptomycin, 10 mM HEPES buffer and 1 mM sodium pyruvate. At the end of incubation media was discarded and triplicates of 200,000 peripheral blood mononuclear cells (PBMCs) were grown in the presence or absence of N peptide pool at 1.5 μg/ml final concentration in 100 μl of R-10 media. The peptide mix covers the whole sequence of the nucleocapsid phosphoprotein. After 16 h of incubation at 37°C, the ELISpot plate was washed followed by incubation with 50 μl biotinylated mouse anti-human IFNγ monoclonal antibody 7-B6-1 diluted to 0.5 μg/ml in 0.5% BSA/PBS for 3 h. Captured IFNγ was detected with 50 μl of anti-biotin monoclonal antibody, diluted 1:750 ml in 0.5% BSA/PBS. After 2 h, plate was washed,

50 µl of nitro blue tetrazolium/5-bromo-4-chloro-3-indolyl-phosphate was added; purple spots appeared within 10 min. Spot numbers were analysed by an ELISpot reader. Frequencies of Cov-2 Spike-specific IFNγ producing cells were calculated by subtracting the number of detected spots in the unstimulated sample from the number of spots detected in the presence of PepTivator SARS-CoV-2 protein N peptide pool (average of triplicates) and were given as IFNγ spot forming cells (SFC)/$1 \times 10^6$ PBMC.

### Flow cytometry

Electroporated fluorescent antibody (IgG-A488) was detected using an Eclipse EC800 flow cytometer after fixing cells with 4% formaldehyde at 1 h post-electroporation. Flow cytometry data were analysed using Eclipse software to obtain mean fluorescence intensity for the entire cell population as well as % cells IgG-A488 positive by gating against cells electroporated with just PBS.

### Microscopy

Cells electroporated with IgG-A488 were washed in fresh media to remove undelivered antibody and plated on 8-well chambered coverslips. Cellular IgG-A488 was detected using a Lumascope LS720 widefield microscope equipped with a $40 \times 0.95$ NA air objective, housed within a 37°C, 5% $CO_2$ humidified incubator.

### Statistical analysis

Unless otherwise indicated, statistical analyses were performed using GraphPad Prism 9 software (GraphPad) employing one-way and two-way ANOVA and least squares fits. Error bars depict the mean ± SEM unless indicated otherwise.

## Data availability

This study includes no data deposited in external repositories.

**Expanded View** for this article is available online.

## Acknowledgements

We are grateful to Dr. Peter Rottier of Utrecht University for reagents and advice on MHV-A59. This work was supported by MRC U105181010 (LCJ), Wellcome Trust Investigator Award 200594/Z/16/Z (LCJ), Wellcome Trust Collaborator Award 214344/A/18/Z (LCJ).

## Author contributions

Conceptualization: LCJ. Methodology: LCJ, AA, DC, MV, TR, DMF, HEB, SLC. Investigation: AA, DC, MV, TR, DMF, HEB. Analysis: LCJ, AA, DC, MV. Writing – original draft: LCJ, AA, DC. Writing – review and editing: LCJ, AA, DC, MV, TR, DMF, HEB.

## Conflict of interest

The authors declare that they have no conflict of interest.

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
