## [Review Process File · The EMBO Journal]

A functional assay for SARS-CoV-2 N-antibodies

Anna Albecka, Dean Clift, Marina Vaysburd, Tyler Rhinesmith, Sarah Caddy, Helen Baxendale, David Favara, and Leo James

DOI: [10.15252/embj.2021108588](https://doi.org/10.15252/embj.2021108588)

Corresponding author(s): Leo James (lcj@mrc-lmb.cam.ac.uk)

Review Timeline:

Submission Date:	27th Apr 21
Editorial Decision:	4th Jun 21
Revision Received:	17th Jun 21
Editorial Decision:	25th Jun 21
Revision Received:	29th Jun 21
Accepted:	2nd Jul 21

Editor: Karin Dumstrei

Transaction Report:

Dear Leo,

Thank you for submitting your manuscript to The EMBO Journal.

Your study has now been seen by two referees and their comments are provided below. As you can see below both referees appreciate the analysis and find it suitable for publication in The EMBO Journal. They raise relative minor concerns that should be fairly easy to sort out. Let me know if we need to discuss anything further - happy to do so via email or video call.

Please also take a look at the attached revision guideline with helpful tips for how to prepare the revision.

I thank you for the opportunity to consider your work for publication.

with best wishes

Karin

Karin Dumstrei, PhD
Senior Editor
The EMBO Journal

The revision must be submitted online within 90 days; please click on the link below to submit the revision online before 2nd Sep 2021.

Referee #1:

In this manuscript the authors develop a new assay to measuring the potential neutralizing activity of N-specific antibodies generated following SARS-CoV-2 infection. The assay involves electroporating the antibodies into a cell and then challenging the cell with virus. This assay allowed the authors to show that when N-specific IgG is electroporated into cells, it is able to prevent viral replication on SARS-CoV-2 in a TRIM21 dependent manner. This gives a functional activity for the high levels of N-IgG observed in sera. They were further able to show that the neutralizing activity correlated with the level of N-IgG in patient sera and that the neutralization was facilitated by N-IgG and not S-IgG. Overall, this is a very interesting study that will be of broad interest as well as having specific relevance to COVID-19 vaccination. The manuscript is clearly written, and the

experiments are carried out to a high standard.

How was it determined that the electroporation has led to the IgG entering the cell?

Is there any evidence that vaccines containing N have better protective capacity than Spike based vaccines? E.g. vaccines based on inactivated viruses such as Sinovac?

Are there any known correlations between N IgG levels and disease severity and/or progression following SARS-CoV-2 infection? This would give biological significance to the functional activity measured.

Do the authors anticipate that all epitopes on N will facilitate neutralization?

Stats for the correlation on Figure 4C are missing.

Referee #2:

In this work, the authors developed an artificial in vitro method called EDNA, by electroporating antibodies into cells, to provide quantitative measurement of N-antibody activity. The author further proved that the N-antibodies neutralize SARS-CoV-2 intracellularly by recruiting the cytosolic Fc-receptor TRIM21. As claimed by the authors, the function of antibodies to intracellular antigens has been shown to prevent infection by arenaviruses, ebolavirus, HCMV, HIV and influenza virus. However, comparing with Spike-induced neutralizing antibodies, Nucleoprotein-antibodies are expected for more in-depth investigation, I recommend acceptance with minor revisions.

Specific comments:

1. The authors are suggested to add an abstract graphic as Figure 1A or Figure S1A, to better summarize their design of this new method.
2. In Figure 1A, the titrations of anti-MHV polyclonal serum added either extracellular or intracellular lack data from 16h to 24h, is there any difficulty quantifying cell area?
3. Line 94, "electroporated L929 cells with serial dilutions of an anti-N monoclonal antibody". Is there any difference among N-antibodies with different epitopes?
4. In Figure 1I. The authors utilized the CHX model to test N-protein levels, and should also present the protein level without electroporation to rule out other perturbations and be a positive control.
5. The "N-antibody" mentioned in the manuscript is ambiguous. It is important to address neutralizing and non-neutralizing functions. Line 50, some references the authors provide are involved non-neutralizing antibodies, are the mechanisms underlying via ADIN pathways?
6. A cohort of four SARS-CoV-2 convalescents is relatively small and inadequate to draw a distinct conclusion of antibody portfolios. The authors may indicate more details in Figure 3A and 3B explanation.
7. Line 141, "... to make it permissive for SARS CoV-2 entry", should it be "sensitive"? not "permissive".
8. According to previous studies and common sense, the antibodies are not efficiently taken up by cells in vitro, however, in this artificial method, this process is bypassed by using electroporation. Since the situation is much complicated in COVID-19 convalescent patients, the authors are suggested adding some sentences to address the limitations of this method in the discussion section.

Referee #1:

Q1. How was it determined that the electroporation has led to the IgG entering the cell?

A1. We established conditions for efficient electroporation of IgGs into cells using Alexa488-labelled antibodies and a combination of flow cytometry and confocal microscopy. In Figure S1H, we show dose-dependent delivery of Alexa488-IgGs by quantifying the mean fluorescence intensity of electroporated cells. In Figure S1I, confocal microscopy images show that Alexa488-IgGs are distributed diffusely throughout the cell 6 hours post-electroporation.

Q2. Is there any evidence that vaccines containing N have better protective capacity than Spike based vaccines? E.g. vaccines based on inactivated viruses such as sinovac?

A2. This is an important question that needs to be addressed but as yet there is very limited data. It is hard to compare existing vaccines, even those just using Spike, because different platforms and delivery modalities are being used. However, all vaccines that have achieved WHO Emergency Use Listing are effective in preventing disease and hospitalization due to COVID-19 (<https://www.who.int/news-room/feature-stories/detail/the-sinovac-covid-19-vaccine-what-you-need-to-know>). Whether there are differences in longevity of protection or efficacy against the continuing emergence of variants remains to be determined.

Q3. Are there any known correlations between N IgG levels and disease severity and/or progression following SARS-CoV-2 infection? This would give biological significance to the functional activity measured.

A3. This is also a crucial question that we are actively seeking to address. We are currently participating in a multi-centre study to address what aspects of humoral immunity correlate with protection. Our first pilot data set has recently been deposited on MedRxiv (<https://www.medrxiv.org/content/10.1101/2021.05.21.21257572v1>). In this dataset, we observed that severe COVID-19 patients had significantly higher N IgG levels and intracellular neutralizing titres than mild or asymptomatic seropositive participants. This is consistent with the high and persistent viral replication and antigen expression in COVID-19 patients. This suggests that N antibody levels are biologically significant and we are seeking to test this further in a larger patient group. Whether this increased antibody response contributes to antibody-mediated immunopathology will also be important to determine.

Q4. Do the authors anticipate that all epitopes on N will facilitate neutralization?

A4. Our previous work on the mechanism of TRIM21-mediated neutralization suggests that the principle factors required for activity are antibody:antigen (<https://pubmed.ncbi.nlm.nih.gov/27881870/>) and antibody:TRIM21 affinity (<https://pubmed.ncbi.nlm.nih.gov/26962230/>). More specifically, off-rate is crucial. We don't anticipate that different epitopes per se will elicit different neutralization activity, unless upon antibody binding the Fc is not accessible for TRIM21 binding. This may be interesting to test in future using a panel of monoclonal antibodies against defined epitopes. We have added a sentence to the discussion to address this question (Lines 267-269).

Q5. Stats for the correlation on Figure 4C are missing.

A5. We have added an inset panel to Figure 4C showing a nonlinear fit to the log transformed data and the R^2 value.

Referee #2:

Q1. The authors are suggested to add an abstract graphic as Figure 1A or Figure S1A, to better summarize their design of this new method.

A1. We are grateful for this suggestion and have added a graphical abstract giving an overview of the new method.

Q2. In Figure 1A, the titrations of anti-MHV polyclonal serum added either extracellular or intracellular lack data from 16h to 24h, is there any difficulty quantifying cell area?

A2. No scans were taken during that time due to instrument availability. We apologise that this is not ideal but are confident in the trajectory of the collected data.

Q3. Line 94, "electroporated L929 cells with serial dilutions of an anti-N monoclonal antibody". Is there any difference among N-antibodies with different epitopes?

A3. Unfortunately, we were not able to source additional monoclonal antibodies. Most described MHV anti-N antibodies were made in the 1980s and have been difficult to get hold of. We are grateful to Dr. Peter Rottier who was able to help us obtain the MHV antiserum we used in our study. However, we believe that different epitopes will be less important than the antibody-antigen off-rate (please see response A4 to Referee 1 above).

Q4. In Figure 1I. The authors utilized the CHX model to test N-protein levels, and should also present the protein level without electroporation to rule out other perturbations and be a positive control.

A4. As requested, we have now included data showing electroporation does not alter N-protein levels.

Q5. The "N-antibody" mentioned in the manuscript is ambiguous. It is important to address neutralizing and non-neutralizing functions. Line 50, some references the authors provide are involved non-neutralizing antibodies, are the mechanisms underlying via ADIN pathways?

A5. We have added the following to the introduction:

Lines X: "Crucially however, because internal antigens are usually hidden inside the virion, antibodies against them do not bind infectious viral particles. Consequently, N-antibodies and similar typically do not block infectious entry of viruses into cells in standard in vitro assays and are described as 'non-neutralizing'. The mechanisms behind the immune protection provided by non-neutralizing antibodies like N-antibodies remain largely unknown."

And to the discussion:

Lines 273-278: "Infection with most enveloped viruses results in high titre antibodies against internal antigens and while these are typically non-neutralizing in vitro (at least as characterized in standard assays) they are often highly protective in vivo[31]. Based on our results here, and recently published{Caddy, 2020 #2507}, we propose that ADIN may provide one underlying mechanism by which non-neutralizing antibodies mediate protection. However, ADIN does not explain how and where non-neutralizing antibodies like those against N protein meet their antigen."

Q6. A cohort of four SARS-CoV-2 convalescents is relatively small and inadequate to draw a distinct conclusion of antibody portfolios. The authors may indicate more details in Figure 3A and 3B explanation.

A6. We have added the following to the results:

Lines 167-169: "The data shows that a good dynamic range in antigen reactivity is capable of being measured but a larger cohort would be required to draw any conclusions about antibody portfolios."
Analysis of the data for correlations in antigen reactivity is performed only on the larger cohort in Figure 4. We are also involved in studies attempting to correlate antibody portfolios with disease severity and progression (see answer A3 to Review 1).

Q7. Line 141, "... to make it permissive for SARS CoV-2 entry", should it be "sensitive"? not "permissive".

A7. We have changed the text as suggested.

Q8. According to previous studies and common sense, the antibodies are not efficiently taken up by cells in vitro, however, in this artificial method, this process is bypassed by using electroporation. Since the situation is much complicated in COVID-19 convalescent patients, the authors are suggested adding some sentences to address the limitations of this method in the discussion section.

A8. We have expanded the discussion section as follows:

Lines 277-281: "However, ADIN does not explain how and where non-neutralizing antibodies like those against N protein meet their antigen. EDNA is an in vitro assay in which antibodies are artificially delivered directly into the cytosol. This is a limitation of the method and does not address the issue of cellular uptake and cytosolic import during natural SARS-CoV-2 infection in vivo."

Dear Leo,

Thank you for submitting your revised manuscript to The EMBO Journal. I have looked at your response and all looks good. I am therefore very pleased to let you know that we will accept the manuscript for publication here. Before sending you the formal acceptance letter there are just a few formatting issues that we need to resolve.

- Please upload individual figures files.
- Please add 3-5 keywords
- Please re-label "Data and materials availability" as "Data Availability" and place this section after Structured Methods. As far as I can see no data is generated that needs to be deposited in a database. If this is correct please state: This study includes no data deposited in external repositories.
- Competing interests should be re-labeled as Conflict of Interest
- The reference format should be alphabetical, 10 et al.
- The movie should be called "Movie EV1". Please also correct callout in text. The legend should be removed from appendix and zipped to the movie file.
- The appendix needs a ToC. Figures in the appendix should be labeled as "Appendix Figure S1". Please also correct callouts in text. Alternatively, you can also upload the appendix figures as Expanded view figures. Please see guide to authors
<https://www.embopress.org/page/journal/14602075/authorguide#expandedview>
- We encourage the publication of source data, particularly for electrophoretic gels and blots, with the aim of making primary data more accessible and transparent to the reader. It would be great if you could provide me with a PDF file per figure that contains the original, uncropped and unprocessed scans of all or key gels used in the figure? The PDF files should be labelled with the appropriate figure/panel number, and should have molecular weight markers; further annotation could be useful but is not essential. The PDF files will be published online with the article as supplementary "Source Data" files.
- We include a synopsis of the paper (see <http://emboj.embopress.org/>). Please provide me with a general summary statement and 3-5 bullet points that capture the key findings of the paper.
- We also need a summary figure for the synopsis. The size should be 550 wide by [200-400] high (pixels). You can also use something from the figures if that is easier.
- Our publisher have done their pre-publication checks on the manuscript and made comments in the figure legends. Please take a look at the "Data Edited Manuscript file" word document to see their marked changes.
- Remove Reagents & Tools Table from MS file and upld as a separate file.

- Rename Methods & Protocols => should be Structured Methods

- Please merge acknowledgements it is now in two parts.

That should be all.

You can use the link below to upload the revised version

With best wishes

Karin

Karin Dumstrei, PhD
Senior Editor
The EMBO Journal

Guide For Authors: <https://www.embopress.org/page/journal/14602075/authorguide>

The revision must be submitted online within 90 days; please click on the link below to submit the revision online before 23rd Sep 2021.

Dear Leo,

Thank you for submitting your revised manuscript to The EMBO Journal. I have now had a chance to take a careful look at it and all looks good.

I am therefore very pleased to accept the manuscript for publication here.

Congratulations on a nice study

With best wishes

Karin

Karin Dumstrei, PhD
Senior Editor
The EMBO Journal

Please note that it is EMBO Journal policy for the transcript of the editorial process (containing referee reports and your response letter) to be published as an online supplement to each paper. If you do NOT want this, you will need to inform the Editorial Office via email immediately. More information is available here: https://emboj.embopress.org/about#Transparent_Process

Your manuscript will be processed for publication in the journal by EMBO Press. Manuscripts in the PDF and electronic editions of The EMBO Journal will be copy edited, and you will be provided with page proofs prior to publication. Please note that supplementary information is not included in the proofs.

Should you be planning a Press Release on your article, please get in contact with embojournal@wiley.com as early as possible, in order to coordinate publication and release dates.

If you have any questions, please do not hesitate to call or email the Editorial Office. Thank you for your contribution to The EMBO Journal.

Corresponding Author Name: Leo James

Journal Submitted to: The EMBO Journal

Manuscript Number: EMBOJ-2021-108588